# Case of Mixed Infection of Toenail Caused by *Candida parapsilosis* and *Exophiala dermatitidis* and In Vitro Effectiveness of Propolis Extract on Mixed Biofilm

**DOI:** 10.3390/jof9050581

**Published:** 2023-05-17

**Authors:** Alana Salvador, Flávia Franco Veiga, Terezinha Inez Estivalet Svidzinski, Melyssa Negri

**Affiliations:** Departamento de Análises Clínicas e Biomedicina, Universidade Estadual de Maringá (UEM), Avenida Colombo, 5790, Maringá CEP 87020-900, PR, Brazil

**Keywords:** biofilm, *Candida parapsilosis*, black yeast, propolis extract

## Abstract

Onychomycosis is a chronic fungal nail infection caused by several filamentous and yeast-like fungi, such as the genus *Candida* spp., of great clinical importance. Black yeasts, such as *Exophiala dermatitidis*, a closely related *Candida* spp. species, also act as opportunistic pathogens. Fungi infectious diseases are affected by organisms organized in biofilm in onychomycosis, making treatment even more difficult. This study aimed to evaluate the in vitro susceptibility profile to propolis extract and the ability to form a simple and mixed biofilm of two yeasts isolated from the same onychomycosis infection. The yeasts isolated from a patient with onychomycosis were identified as *Candida parapsilosis sensu stricto* and *Exophiala dermatitidis*. Both yeasts were able to form simple and mixed (in combination) biofilms. Notably, *C. parapsilosis* prevailed when presented in combination. The susceptibility profile of propolis extract showed action against *E. dermatitidis* and *C. parapsilosis* in planktonic form, but when the yeasts were in mixed biofilm, we only observed action against *E. dermatitidis*, until total eradication.

## 1. Introduction

Onychomycosis is a chronic infection caused by nail fungi, primarily caused by dermatophytes, followed by yeasts and non-dermatophyte filamentous fungi [1]. Yeasts of the genus *Candida* are widely reported to cause various pathologies [2] and commonly cause onychomycosis, with species of the *C. parapsilosis* complex being the main ones involved in nail mycoses [3,4]. The black yeast, *Exophiala dermatitidis*, is characterized as a pathogen that acts opportunistically [5], often present in the respiratory tract of patients with cystic fibrosis and/or causing other types of superficial and skin infections, which are underdiagnosed worldwide [6].

*C. parapsilosis sensu stricto*, part of the human skin microbiota, is commonly isolated from hands and has the ability to form biofilm on different surfaces [3,4,7,8], which can be simple or mixed biofilm [9,10]. *E. dermatitidis* can form biofilms as well [5]. It is common knowledge that infectious diseases are affected by organisms organized in biofilm, especially in onychomycosis [1,11,12,13].

The most common interactions between microorganisms are commensalism and antagonism; in commensalism, one population benefits, and the other remains neutral, whereas antagonism is defined as the death, injury, or inhibition of one microorganism by the other [14]. Pathirana et al. demonstrated that mixed biofilms among *Candida* spp. provide a growth advantage, especially for non-*C. albicans* [15]. Gupta et al. demonstrated the high prevalence of mixed infections in global onychomycosis between dermatophytes and non-dermatophytes [16]. There are reports of onychomycosis also by non-dermatophytes such as *Rhodotorula mucilaginosa* and *Candida parapsilosis* [17], *R. mucilaginosa* and *Trichosporon asahii* [18]. Studies conducted in dishwashers found *Candida* spp. and *Exophiala* spp. grown in association. This finding is very worrisome because it demonstrates that these yeasts, potential causes of fungemia, are present in our daily lives [19,20,21], but as far as is known, there are no reports indicating the association of the two yeasts in human mycoses.

The treatment for onychomycosis is a challenge because while it is common to use systemic antifungal drugs, such as azoles, terbinafine and griseofulvin, these are limited options [1]. The difficulty in treating onychomycosis is common due to the long treatment time (about 12 months), reinfections, toxicity of systemic antifungal drugs or even treatment withdrawal by the patient [22]. Biofilm formation is a worrisome factor, as it can hinder treatment [12].

In this context, natural products are viewed as potential options, as described in the literature [23]. Recent research confirms the efficacy of the use of natural products, such as propolis extract, in the treatment of onychomycosis [1,24,25]. Propolis is a resin produced by bees (*Apis mellifera* L.) from salivary secretions and wax with exudates collected from plants and possesses biological actions such as antioxidant, antinociceptive and anti-inflammatory activity [26]. The use of propolis extract as an antifungal agent can be considered promising, as it has already been described as effective against yeasts of the *Candida* genus and biofilm-related to this genus [1,27]. However, little is known about the effect of propolis on mixed biofilms of *Candida* with other species, let alone on black yeasts [28,29].

Based on the importance of better understanding the etiopathogenesis of onychomycosis caused by yeasts, as well as the need for studies on biofilms associated with this mycosis and the incessant search for new antifungal therapies. The aim of this study was to evaluate the capacity to form simple and mixed biofilm and to verify the action of propolis extract (PE) on biofilm formed by *C. parapsilosis sensu stricto* and *E. dermatitidis* isolated from onychomycosis.

## 2. Materials and Methods

### 2.1. Case Report

A 50-year-old patient was seen at the Teaching and Research Laboratory (LEPAC) with a two-year history of deformity, thickening of the left toenail, with inflammation around the toenail. Physical examination showed that the three layers (dorsal, intermediate, and ventral) of toenail were affected with paronychia (Figure 1A). Six months ago, the patient was treated with oral itraconazole 200 mg twice daily for one week each month for four months but did not respond well to treatment. The patient reported no other illness during this period. Standing Committee on Ethics in Research Involving Human Beings, under registration numbers 615,643 (isolation and identification of microorganisms).

### 2.2. Microorganisms

Samples were collected, isolated and identified according to the classic methodology by direct mycological examination with potassium hydroxide (Sigma-Aldrich, Saint Louis, MO, USA) and 0.5% Evans blue (Sigma-Aldrich, Saint Louis, MO, USA). Samples were also spread on Sabouraud dextrose agar (SDA; Kasvi, Madrid, Sapain) and Mycosel (DifcoTM, Detroit, MI, USA). After isolation, identified yeasts were confirmed by macroscopic and microscopic characteristics, as well as mass spectrometry assisted by flight time desorption/ionization matrix (MALDI TOF-MS) and processed with a Vitek MS mass spectrometer using the Myla or Saramis software for data interpretation [30,31]. For confirmation of the isolated agents, two independent collections were performed with an interval of one month after the first collection, repeating the same isolated agents in both collections. The yeasts were maintained in the Mycotech Laboratory of Medical Mycology of the Universidade Estadual de Maringá in Sabouraud dextrose broth (SDB; Kasvi, Italy) and glycerol at −80 °C. Assays were performed with the yeasts *Candida parapsilosis* and *Exophiala dermatitidis*, both from clinical isolates of the Teaching and Research Laboratory of Clinical Analysis (LEPAC). In each experiment, yeasts were cultured on SDA for 3 days at 35 °C.

### 2.3. Experimental In Vitro Condition

#### 2.3.1. Simple and Mixed Biofilm Formation

For simple biofilm formation, colonies of each type of yeast were scraped after 3 days at 35 °C on SDA, collected with 0.85% of sterile saline and conidia counted on Neubauer’s chamber. The inoculum was adjusted to a final concentration of 1 × 10^5^ conidia/mL in RPMI 1640 medium (Roswell Park Memorial Institute, Gibco, Grand Island, NY, USA) with L-glutamine (without sodium bicarbonate) and 0.165 M (3-(N-morpholino) propanesulfonic acid (pH 7.0) (Sigma-Aldrich, Saint Louis, MO, USA). Finally, 200 μL of this suspension was plated on 96 well microplates.

For mixed biofilm formation, after 3 days at 35 °C on SDA, we proceeded in the same way as mentioned above. Different inoculums were tested from 1 × 10^3^ to 1 × 10^6^ to define the best concentration for the assay (Appendix A) until the 1 × 10^5^ conidia/mL concentration in RPMI buffered with MOPS was chosen in a 1:1 ratio with the microorganisms. Finally, 200 μL of this suspension was plated on 96 well microplates.

Microplates (simple and mixed biofilm) were incubated at 35 °C at 120 rpm per minute for up to 96 h for biofilm formation. Every 24 h, 100 μL of RPMI 1640 medium was removed from each well, and an equal volume of fresh RPMI 1640 medium was added. Biofilms collected at 24, 48, 72 and 96 h were quantified in terms of viability assay, total biomass, and metabolic activity. All experiments were repeated on three separate occasions, with individual samples evaluated in triplicate.

#### 2.3.2. Biofilm Viability Assay by Colony Forming Units (CFU)

After each biofilm formation, the supernatant from each well was removed, and each well was washed for removal of non-adhered fungi. To remove the biofilm, 200 μL of saline was added to each well and vigorously scraped with a tip, transferring the entire volume to a conical tube. This process was repeated 5 times until a total volume of 1000 μL was achieved in the conical tube, which was agitated for 30 s and sonicated with an equipment Sonic Dismembrator Ultrasonic Processor (Fisher Scientific, Waltham, MA, USA) at 30% amplitude for 50 s. Serial dilutions were made in PBS, and aliquots of 15 μL were plated on SDA and incubated at 35 °C for 48 h. After incubation, the grown colonies were visually counted, and the CFU values were standardized according to the well area (Log_10_/cm^2^) [32].

#### 2.3.3. Biofilm Metabolic Activity Assay by XTT Reduction (XTT)

This assay was developed based on the technique previously described by Galletti et al. [25]. The reduction assay of the tetrazolium salt 2,3-(2-methoxy-4-nitro-5-sulphophenyl)-5-([phenylamino]carbonyl)-2H tetrazolium hydroxide (XTT; Sigma-Aldrich, Saint Louis, MO, USA) was used to determine in situ biofilm mitochondrial activity. After biofilm formation, each well was washed for removal of non-adhered fungi. A total of 100 μL of a solution containing XTT, phenazine methosulphate (PMS) and 0.85% sterile saline was added to each well. Sixty microliters of sterile saline were then added, followed by 20 μL of XTT stock solution (500 μg/mL) and 20 μL of PMS stock solution (50 μg/mL). Final concentrations of XTT and PMS in the wells were 100 and 10 μg/mL, respectively. The microplates were then incubated at 35 °C for 3h, protected from light. After incubation, absorbance of the obtained solution was read in a microtiter plate reader at 492 nm and absorbance values were standardized per unit area of the well (absorbance/cm^2^).

#### 2.3.4. Total Biofilm Biomass Quantification by Crystal Violet Staining (CV)

This assay was developed based on the technique previously described by Galletti et al. [25]. After biofilm formation, each well was washed for removal of non-adhered fungi and 200 μL of methanol was added for 15 min to fix the biofilms. Then, 200 μL of crystal violet (1% *v*/*v*) was added, followed by a 5 min incubation at room temperature. Microplates were then washed with sterile distilled water, and 200 μL of acetic acid (33% *v*/*v*) was then added to dissolve the stain. The obtained solution was read in a microtiter plate reader at 620 nm, and absorbance values were standardized per unit area of the well (absorbance/cm^2^).

#### 2.3.5. Propolis Extract (PE) 

The propolis extract used was characterized by Corrêa et al. [33]. Collected from hives located in northern Paraná State, Brazil. It was prepared with a propolis/ethanol ratio of 30/70 (*w*/*w*) by turbo extraction, centrifuged at 3500× *g*, three times for 15 min each, with two 5 min intervals, filtered through filter paper and filled to the initial weight with ethanol. PE was characterized by relative density of 0.8617 ± 0.00 g/mL, pH of 5.17 ± 0.05, dryness residue of 16.11% ± 0.07, and total phenol content (TPC) of 2.68% ± 0.09. All the tests with the propolis extract were performed together with the ethanol (Synth, São Paulo, Brazil) at the corresponding concentration of each test.

#### 2.3.6. Antifungal Activity

In vitro antifungal susceptibility for planktonic cells was performed according to Corrêa et al. [33]. The serial dilution was performed at a ratio of 1:2, and PE concentrations ranged from 13,400 to 26 μg/mL of total phenol content (TPC). Moreover, a test using only ethanol (70%, *w*/*w*) was also accomplished, aiming to serve as a control to detect possible interference of this solvent. After incubation at 35 °C for 48 h, the minimum inhibitory concentration (MIC) was determined by direct observation. The minimum fungicidal concentration (MFC) was determined by seeding the suspensions exposed to different PE concentrations on SDA plates. Plates were incubated at 35 °C for 24 h, and the MFC was defined as the lowest concentration of the test compound in which no recovery of microorganisms was observed. The obtained MIC and MFC results were corresponding.

#### 2.3.7. Antibiofilm Activity

Evaluation of antibiofilm activity was performed according to Veiga et al. and Galletti et al. [1,25]. After simple and mixed biofilm formation, as described above, for 48 h, each well was washed for removal of non-adhered fungi. PE was then added at 1675 μg/mL, which corresponds to twice the MIC found in the assay already described. Microplates were then incubated at 35 °C for 24 h, and antibiofilm activity was determined by MFC.

### 2.4. Statical Analysis

Experiments were performed in three independent trials and in triplicate. Data were expressed as mean ± standard deviation. GraphPad Prism 8.0.2 software (OriginLab Corporation, Northampton, MA, USA) was used for statistical analysis. Data were statistically evaluated by one-way analysis of variance (ANOVA), and post hoc comparisons of the means of individual groups were performed using Tukey’s test. Values of *p* ≤ 0.05 were considered statistically significant.

## 3. Results

### 3.1. Onychomycosis by Candida parapsilosis sensu stricto and Exophiala dermatitidis

The patient’s lesion was localized on the distal end of the nail of the first polydactyl with paronychia and the nail plate affecting its three layers (dorsal, intermediate, and ventral) (Figure 1A). In culture, there was reproducible growth at all three inoculated sites, in all three SDA tubes, and in two independent collections. Mixed growth was presented in the culture medium after 15 days, which showed a yeast appearance and black, turning olive-gray with the development of aerial mycelium with age and growth of other yeast white, creamy, shiny and smooth/rugged colonies (Figure 1B). Subsequently, colonies were isolated and identified according to their morphological and biochemical characteristics and confirmed by MALDI-TOF as *Candida parapsilosis sensu stricto* and *Exophiala dermatitidis*.

### 3.2. Candida parapsilosis and Exophiala dermatitidis from Onychomycosis Were Capable of Biofilm Formation over Time

Both yeasts were able to form biofilms over time (Figure 2). In the evaluation of viable cells, *C. parapsilosis* showed constant viable cell number (24 h with 6.52 Log_10_ UFC/cm^2^; 48 h with 6.49 Log_10_ UFC/cm^2^) until 72 h (6.72 Log_10_ UFC/cm^2^), decreasing significantly (*p* < 0.05) at 96 h (5.04 Log_10_ UFC/cm^2^). The total biomass was higher at 72 h (2.47 Abs_620_/cm^2^) and decreased significantly (*p* < 0.05) at 96 h (1.12 Abs_620_/cm^2^). Mitochondrial metabolic activity, on the other hand, increased significantly (*p* < 0.05) from 72 h (0.69 Abs_492_/cm^2^) to 96 h (0.85 Abs_492_/cm^2^). The number of viable cells of *E. dermatitidis* was more heterogeneous, increasing significantly (*p* < 0.05) from 24 h (5.51 Log_10_ UFC/cm^2^) to 48 h (6.49 Log_10_ UFC/cm^2^), decreasing at 72 h (6.07 Log_10_ UFC/cm^2^) and increasing significantly again (*p* < 0.05) at 96 h (6.59 Log_10_ UFC/cm^2^). The mitochondrial metabolic activity and total biomass of the biofilms were similar to *C. parapsilosis*.

### 3.3. Candida Parapsilosis Is Benefited in the Mixed Biofilm with Exophiala dermatitidis

When performing the mixed biofilm formation of the two yeasts, it was possible to observe that together they were also able to form biofilm over time (Figure 3). The number of viable cells (Figure 3A) was highest at 24 h (9.57 Log_10_ UFC/cm^2^) (Appendix A), where both showed similar growth (5.23 Log_10_ UFC/cm^2^
*Candida parapsilosis* and 4.36 Log_10_ UFC/cm^2^
*Exophiala dermatitidis*). However, over time the number of viable cells of *C. parapsilosis* (48 h with 4.44 Log_10_ UFC/cm^2^; 72 h with 4.75 Log_10_ UFC/cm^2^; 96 h with 4.51 Log_10_ UFC/cm^2^) was higher than *E. dermatitidis* (48 h with 3.30 Log_10_ UFC/cm^2^; 72 with 3.77 Log_10_ UFC/cm^2^; 96 h with 4.00 Log_10_ UFC/cm^2^). The highest mitochondrial activity (Figure 3B) was at 72 h (0.84 Abs_492_/cm^2^), decreasing significantly (*p* < 0.05) at 96 h (0.58 Abs_492_/cm^2^). The total biomass of the mixed biofilm (Figure 3C) was significantly higher (*p* < 0.05) at 72 h (3.41 Abs_620_/cm^2^), with a significant decrease (*p* < 0.05) at 96 h (0.35 Abs_620_/cm^2^).

### 3.4. Propolis Extract Has an Action on Planktonic Cells of C. parapsilosis and E. dermatitidis, and Reduces the Simple Biofilm of These Yeasts

When performing the susceptibility test of planktonic cells to PE, the MIC and MFC of *C. parapsilosis sensu stricto* and *E. dermatitidis* were at the concentration of 837 μg/mL, as demonstrated in Figure 4. When performing the susceptibility test on the 48 h formed biofilm, PE was able to reduce the simple biofilm of both yeasts significantly (*p* < 0.05). Single biofilm of *C. parapsilosis* in the absence of PE showed 6.50 Log_10_ UFC/cm^2^ and 6.25 Log_10_ UFC/cm^2^ when treated with PE, a reduction of 4%. *E. dermatitidis* showed 6.49 Log_10_ UFC/cm^2^ without PE and 5.49 Log_10_ UFC/cm^2^ when treated with PE, a reduction of 15% (Figure 5A). The solvent at the concentrations used had no activity on the fungus.

### 3.5. Exophiala dermatitidis Is Inhibited in Mixed Biofilm in the Presence of Propolis Extract

In the mixed biofilm (Figure 5B), it was possible to observe *E. dermatitidis* is inhibited in the presence of propolis, as there was no growth of this yeast (100% reduction), while *C. parapsilosis* showed a 14% reduction (6.87 Log_10_ UFC/cm^2^ without PE and 5.96 Log_10_ UFC/cm^2^ when treated with PE).

## 4. Discussion

Although most onychomycosis is caused by dermatophytes, when a dermatophyte grows concomitantly with another fungus in culture, either microbiota or environmental, it is usually considered a contaminant [34]. The identification of mixed onychomycosis is a recent finding with high prevalence. Indeed, with the advancement of molecular techniques used for identification in addition to classical methodology, it is now possible to identify the organisms of infections more accurately [16]. Thus, this case caught our attention because it is the first report of isolation of both yeasts, *Candida parapsilosis* and *Exophiala dermatitidis*, causing onychomycosis. Moreover, we evaluated the capacity of simple and mixed biofilm formation in vitro, as well as the resistance of these yeasts against a natural product, the extract of propolis.

To determine if two organisms are indeed causing onychomycosis, reproducibility of colony growth from the same collected sample is necessary [18]. In this case, the material collected from the nail showed scarce fungal structures in the direct mycological examination, and the culture in SDA showed growth of two yeasts concomitantly and reproducibly. The identification method by MALDI-TOF mass spectrometry was then performed and confirmed *Candida parapsilosis sensu stricto* and *Exophiala dermatitidis*. Although *C. parapsilosis* has long been known as a common contaminant in clinical nail specimens, it is currently emerging as one of the important etiological agents of onychomycosis in association with other fungal pathogens or alone [2,13,16,17]. On the other hand, the environmental fungus *E. dermatitidis* should not be neglected when they are isolated from clinical specimens. This agent is causative of skin and subcutaneous tissue infections and systemic infections but rarely causes onychomycosis [2,5,6,19,20,21,35]. Thus, we are reporting a rare case of mixed infection of the toenail caused by *C. parapsilosis* and *E. dermatitidis*. Possibly *E. dermatitidis* benefits from this mixed biofilm situation and can infect and remain in the host causing the infectious process.

Biofilm is known to play an important role in the pathogenesis of onychomycosis and may be a possible cause of the persistence of infection and resistance to antifungal therapies [13,36]. Furthermore, the biofilm consists of a complex and dynamic architecture that presents several challenges for its in vitro characterization. There are several direct and indirect methods that have been used to quantify cells in biofilms [37]. Direct counting methods allow the counting of cells that can be cultured, including plate counts. Indirect measurement methods include determining total biomass by crystal violet assay and quantifying cell viability by tetrazolium salts, for example [25,32,37]. It should be noted that these methods, both direct and indirect, are complementary as they provide distinct information about biofilm characteristics and architecture [37].

When performing simple biofilm formation, we observed that yeasts are able to form biofilms in vitro evaluated by CFU, crystal violet and XTT method. *C. parapsilosis sensu stricto* is known for its ability to form a biofilm, and Modiri et al. [38] showed that *C. parapsilosis sensu stricto* was the highest producer of biofilm compared to other species of the complex. In this same study, it was observed that *C. parapsilosis sensu stricto* produced the maximum amount of biofilm at 72 h evaluated by biomass, corroborating our results.

*E. dermatitidis* also showed the ability to form a biofilm, with the highest amount of total biomass in 72 h. A study by Kirchhoff et al. [39] demonstrated the ability of this black yeast to form biofilm and increase the amount of biomass with time. Moreover, this same study showed that the total biomass and the metabolic activity of the biofilm are not linearly associated. The main disadvantage of the crystal violet assay is its non-specific nature, in that it does not distinguish between living and dead cells, hyphae and extracellular matrix [25,40]. On the other hand, tetrazolium salts are one of the most widely used tools in biology for real-time monitoring of cell viability and metabolism in vitro; however, the assessment of metabolic activity can be influenced by the structure of the extracellular matrix and the metabolism of the yeast within the biofilm; thus the two methods are complementary to evaluate the biofilm [25,37,40]. The results of the metabolic activity evaluated by XTT showed a decrease in 72 h, which can be explained by the presence of melanin in the yeast cell wall, as well as the fact that a thick extracellular matrix can reduce the diffusion of oxygen and nutrients and thus reduce the metabolic activity detected [39].

Then, we performed the mixed biofilm formation between *C. parapsilosis* and *E. dermatitidis* in vitro. Although we evaluated the ability to form biofilm under in vitro conditions rather than ex vivo in vivo, we observed that at 24 h, the CFU of both yeasts were equivalent, but at later times the number of *E. dermatitidis* was lower than *C. parapsilosis*. A study by Pathirana et al. [15] showed that some biofilms of non-*Candida albicans* benefit when in mixed biofilm with *Candida albicans*. Added to the etiopathogenesis of onychomycosis with fungal biofilm and the increasing incidence of mixed infection causing onychomycosis, there are increasing reports of unusual mixed infection between dermatophytes and non-dermatophyte fungi as well as non-dermatophyte fungi with other non-dermatophyte fungi [13,16,22]. The relation between *C. parapsilosis* and *E. dermatitidis* is not reported in the literature. Onychomycosis by *E. dermatitidis* is rare; possibly, this black yeast benefits from this mixed biofilm situation and can infect and remain in the host causing the infectious process. Meanwhile, Taj-Aldeen et al. [41] show a possible association between *C. krusei* causing fungemia and *E. dermatitidis* colonizing the pulmonary tract in a cancer patient, where probably, *E. dermatitidis* also played a role in the morbidity of the case. The dimorphic character of black yeast is common knowledge, changing from the yeast state to the hyphae state, which is part of the biofilm formation process that may contribute to this association with other yeast [39,42].

The susceptibility test to propolis extract demonstrated that the concentration of 837 μg/mL resulted in the absence of growth of the yeasts in the planktonic state. A study by Tobaldini-Valerio et al. [32] demonstrated that *C. parapsilosis* was susceptible to PE with a MIC range of 220 to 880 μg/mL of TPC, corroborating our results. However, to date, no results have been found for the use of PE in *E. dermatitidis*.

Regarding the effect of PE on the simple biofilm of *C. parapsilosis* and *E. dermatitidis*, for both yeasts, the reduction of biofilm was significant (*p* < 0.05) when PE was added. For *E. dermatitidis*, the reduction was 1 Log, corroborating the literature in which PE is able to reduce single fungal biofilm, the concentration varying according to the species and type of propolis [25,32,43].

When performing the susceptibility test to PE at the same concentration performed for the simple biofilms, we observed in the mixed biofilm. The yeast *C. parapsilosis* showed a reduction, while *E. dermatitidis* was eradicated in the mixed biofilm and in the presence of PE. Mixed fungal biofilms harbor more than one fungal species where exchanges that potentiate the effects of these virulence factors can occur, with benefits for all species or only for some species [42,44]. Regarding the effect of propolis on mixed biofilms, research shows that this natural product is also able to reduce mixed [45,46]. Although there are many studies to evaluate the effect of propolis against causative organisms of onychomycosis, such as dermatophyte fungi, yeasts of the genus *Candida*, and non-dermatophyte filamentous fungi [25,32,43,47,48,49,50,51], there is little research on propolis on mixed fungal biofilms. However, it is important to note that PE has shown efficacy in the treatment of onychomycoses [1,24,52,53].

## Figures and Tables

**Figure 1 jof-09-00581-f001:**
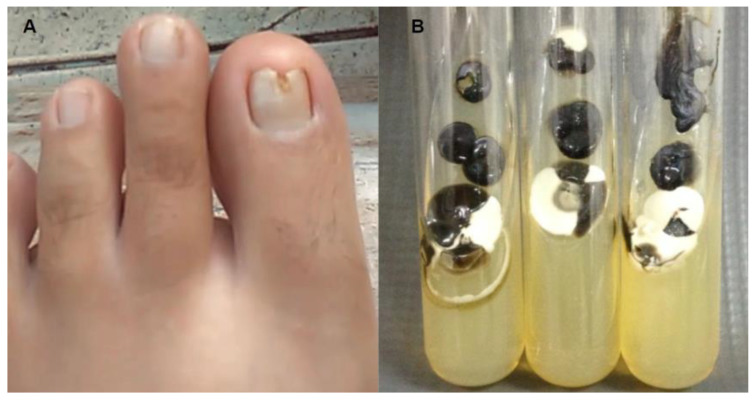
(**A**) Patient with onychomycosis on the toenail; (**B**) Isolation and growth of *Candida parapsilosis* and *Exophiala dermatitidis* reproductively on SDA medium.

**Figure 2 jof-09-00581-f002:**
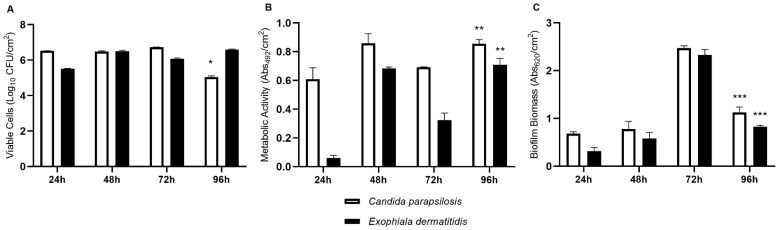
In vitro simple biofilm formation over time of *Candida parapsilosis* and *Exophiala dermatitidis* with an initial concentration of 1 × 10^5^ conidia/mL in RPMI 1640 medium, from 24 h to 96 h. (**A**) Biofilm viability assay by colony forming units (CFU); (**B**) Biofilm metabolic activity by XTT reduction. Absorbance values in these assays were determined at 492 nm; (**C**) Biofilm biomass quantification by crystal violet staining. Absorbance values in these assays were determined at 620 nm. * statistical difference is significantly lower than 24, 48 and 72 h (*p* < 0.05); ** statistically significant increase compared to 72 h (*p* < 0.05); *** statistically significant reduction compared to 72 h (*p* < 0.05).

**Figure 3 jof-09-00581-f003:**
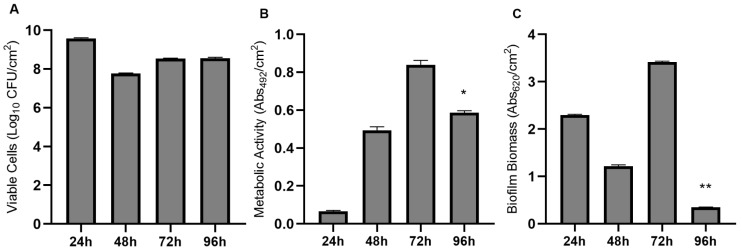
In vitro mixed biofilm formation over time between *Candida parapsilosis* and *Exophiala dermatitidis* with an initial concentration of 1 × 10^5^ conidia/mL in RPMI 1640 medium. (**A**) Biofilm viability assay by colony forming units (CFU); (**B**) Biofilm metabolic activity by XTT reduction. Absorbance values in these assays were determined at 492 nm; (**C**) Biofilm biomass quantification by crystal violet staining. Absorbance values in these assays were determined at 620 nm. * statistically significant reduction compared to 72 h (*p* < 0.05). ** statistically significant reduction compared to 72 h (*p* < 0.05).

**Figure 4 jof-09-00581-f004:**
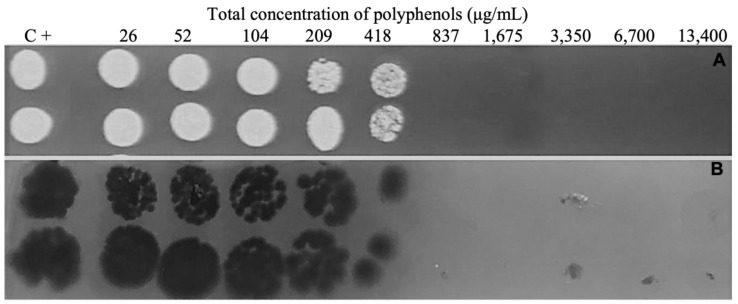
Evaluation of in vitro susceptibility of propolis extract on planktonic cells of *Candida parapsilosis* (**A**) and *Exophiala dermatitidis* (**B**). Evaluated through minimum inhibitory concentration (MIC) and minimum fungicidal concentration (MFC), defined as the ability to inhibit the evaluated fungal population. MIC and MFC correspond to the same values; C + corresponds to the growth of yeasts in the absence of propolis extract.

**Figure 5 jof-09-00581-f005:**
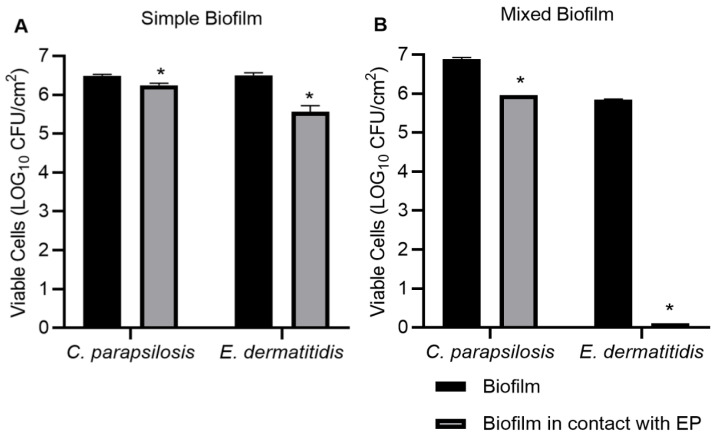
In vitro evaluation of susceptibility of propolis extract on simple (**A**) and mixed (**B**) biofilm formed after 48 h. * Statistically significant reduction compared to biofilm formed in the absence of propolis extract (*p* < 0.05).

## Data Availability

Not applicable.

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
