# Peer review of "Case of Mixed Infection of Toenail Caused by *Candida parapsilosis* and *Exophiala dermatitidis* and In Vitro Effectiveness of Propolis Extract on Mixed Biofilm"

_jof, 2023, doi:10.3390/jof9050581_

Round 1
Reviewer 1 Report
Introduction
Line 55 change the word “promising”
Elaborate on the role of both two Fungi in the etiology of the onychomycosis
Agents – you mean microbes/fungi? Change here and elsewhere in text
M & M
Allocate a specific paragraph on the propolis you are using. For example: where obtained from? Purity? Solvent? Stability?? Concentration used? Ready-made? Storage?
Do you have controls of the solvents in all of the experiments?
Add the MS / MALDI spectrum to supplementary data.
Use the term like homo specie rather that simple biofilm and instead of mixed biofilm use term hetero species biofilm
Line 115 elaborate on 30% amplitude?
Line 116 how were the microbes plated? What dilutions? How many repetitions? Which technique of spreading on agar? Counted under binocular?
Line 155 CIM? CFM?
Line 164 CFM?
Is a sample from one patient indicative? May be collect from many more patients?
Change titles 3.5 & 3.3 and also in text: “favored, disadvantage, disfavor”
Results:
How pure is your isolating and purification of the two fungi?
Section 3.3: elaborate in Methods how you distinguished between the two fungi in experimental set up of biofilm of both fungi. An accurate way of distinguishing between microbes is the qPCR technique, using for example specific 18 S as indicative for each microbial strain.
Fig. 2 why you observed flocculation in the results? discuss in discussion section
Fig. 3 same as above
Fig. 2 compare the results between the A, B, C graphs – maybe in discussion
Fig. 3 same as above
2.2.5 why a description of propolis in this section? Is it different propolis then the rest of the study? See my remark above on propolis
3.3 is a mixed biofilm with no PE? How did you check the presence of each fungi? Why this kinetics graph important here?
Please check again the percentage of reduction especially in the log scale in graphs
3.5 Fig 5 B describes the effect of PE on both fungi in a mixed biofilm. How this procedure was conducted?
In all graphs you need control of the solvent of the PE
Line 305 EP?
Maybe add a picture of the biofilm/s
Discussion
Explain why these two fungi where chosen especially the exophiala. Are they the main ethiological fungi in onychomycosis? This section in the discussion is not that convincing
Compare your results to other fungi as: Candida, Trichophyton
Compare the growth of the mixed biofilm with / without PE
The ability of the fungi to form biofilm should be understood in text that it is in microplates dishes not on biotics surfaces. The support surfaces in vivo are of quite different in chemical texture etc than your model.
Line 305 and 307: EP?
Line 271 change agents to microbes
271- on this section “to determine if the two agents are indeed causing onychomycosis” is not convincing. Use a biotic model.
You should aim the study at either proving that the two fungi are important etiological factors in onychomycosis or proving an effect of PE on fungi associated with the disease.
Author Response
Reviewer 1
Introduction
Line 55 change the word “promising”
Answer: Thank you for the great suggestion, we change in the final version of the manuscript.
Elaborate on the role of both two Fungi in the etiology of the onychomycosis
Answer: Thank you for the great suggestion, unfortunately there are no reports in the literature so far about onychomycosis by both yeasts. So, we found no studies clearly demonstrating this mechanism. It has been described that these yeasts are opportunistic agents, especially in immunocompromised patients, and is unusual in onychomycosis by Exophiala dermatitidis.
Agents – you mean microbes/fungi? Change here and elsewhere in text
Answer: Thank you for the great suggestion, we change in the final version of the manuscript.
M & M
Allocate a specific paragraph on the propolis you are using. For example: where obtained from? Purity? Solvent? Stability?? Concentration used? Ready-made? Storage?
Answer: Thank you for the great suggestion, we allocate a specific paragraph about the propolis extract used.
Do you have controls of the solvents in all of the experiments?
Answer: Yes, we always perform solvent control in all experiments. However, since there was no influence of the solvent on the concentrations used, we do not present the values. In the final version of the manuscript we included the sentence in the M & M "all the tests with the propolis extract were performed together with the solvent at the corresponding concentration of each test" and in the results "The solvent at the concentrations used had no activity on the fungus".
Add the MS / MALDI spectrum to supplementary data.
Answer: For the MALDI TOF-MS method, the yeasts were prepared and processed with a Vitek MS mass spectrometer using the Myla or Saramis software for data interpretation. Thank you for the great suggestion, we added this information to the final version of the manuscript.
Use the term like homo specie rather that simple biofilm and instead of mixed biofilm use term hetero species biofilm
Answer: Dear reviewer, thank you for your suggestion. However, we have chosen to follow the denomination "simple biofilm" and "mixed biofilm" according to another article by our research group Jarros et al., 2022 and according to Gupta et al., 2020, in which co-infections are called mixed infections.
Jarros, I.C.; Barros, I.L.E.; Prado, A.; Corrêa, J.L.; Malacrida, A.M.; Negri, M.; Svidzinski, T.I.E. Rhodotorula Sp. and Trichosporon Sp. Are More Virulent After a Mixed Biofilm. Mycopathologia 2022, 187, 85–93.
Gupta, A.K.; Taborda, V.B.A.; Taborda, P.R.O.; Shemer, A.; Summerbell, R.C.; Nakrieko, K.-A. High Prevalence of Mixed Infections in Global Onychomycosis. PLoS One 2020, 15, e0239648.
Line 115 elaborate on 30% amplitude?
Answer: In sonication we use the equipment Sonic Dismembrator Ultrasonic Processor (Fisher Scientific), the amplitude is the intensity of the vibration of the probe tip. The amplitude used for our study was 30%.
Line 116 how were the microbes plated? What dilutions? How many repetitions? Which technique of spreading on agar? Counted under binocular?
Answer: Biofilms were analyzed by counting colony forming units (CFU) determination. Serial dilutions were made in PBS, aliquots of 15 μL were plated on SDA and incubated for 48 h at 35°C. After incubation, the grown colonies were visually counted and the CFU values were standardized according to the well area (Log10/cm2). This information was added in the final version of the manuscript.
Line 155 CIM? CFM?
Answer: A mistake occurred in the use of the abbreviation. We change in the final version of the manuscript.
Line 164 CFM?
Answer: A mistake occurred in the use of the abbreviation. We change in the final version of the manuscript.
Is a sample from one patient indicative? May be collect from many more patients?
Answer: Yes, this case happened in a single patient. This case caught our attention because it is the first report in the literature of isolation of both yeasts, Candida parapsilosis and Exophiala dermatitidis causing onychomycosis. However, the main objective of the study was to evaluate the in vitro susceptibility profile to propolis extract and the ability to form simple and mixed biofilm of two yeasts isolated from the same onychomycosis infection obtained from a patient sample.
Change titles 3.5 & 3.3 and also in text: “favored, disadvantage, disfavor”
Answer: Thank you for the great suggestion, we change in the final version of the manuscript.
Results:
How pure is your isolating and purification of the two fungi?
Answer: As shown in Figure 1, the yeasts are distinct phenotypically, where E. dermatitidis is a melanized yeast, presenting its colony black. Whereas C. parapsilosis is a hyaline yeast, presenting its colony white. Thus, it is easy to visually distinguish between the isolates.
Section 3.3: elaborate in Methods how you distinguished between the two fungi in experimental set up of biofilm of both fungi. An accurate way of distinguishing between microbes is the qPCR technique, using for example specific 18 S as indicative for each microbial strain.
Answer: Thanks for the suggestion and excellent methodology. However, as mentioned in the previous answer, the yeasts are distinct phenotypically, where E. dermatitidis is a melanized yeast, presenting its colony black. C. parapsilosis is a hyaline yeast, with a white colony. Thus, it is easy to visually distinguish between the isolates.
Fig. 2 why you observed flocculation in the results? discuss in discussion section
Fig. 3 same as above
Fig. 2 compare the results between the A, B, C graphs – maybe in discussion
Fig. 3 same as above
Answer: Dear reviewer, thank you for your suggestion. The observed variations are already commented on in the discussion and we have inserted some more information in the new version.
2.2.5 why a description of propolis in this section? Is it different propolis then the rest of the study? See my remark above on propolis
Answer: We allocate a specific paragraph about the propolis extract used. Your contribution was very important, so we allocate at one paragraph, aiming to avoid ambiguities.
3.3 is a mixed biofilm with no PE? How did you check the presence of each fungi? Why this kinetics graph important here?
Answer: Yes it is a mixed biofilm without the PE. As mentioned in the previous answer, the yeasts are distinct phenotypically, where E. dermatitidis is a melanized yeast, presenting its colony black. C. parapsilosis is a hyaline yeast, with a white colony. Thus, it is easy to visually distinguish between the isolates. The kinetics is important to show that the yeasts together are able to form biofilm and that there are differences from the mixed biofilm to the simple biofilm.
Please check again the percentage of reduction especially in the log scale in graphs
Answer: Dear reviewer, thank you for your suggestion. We checked the results and we have inserted more information in the new version.
3.5 Fig 5 B describes the effect of PE on both fungi in a mixed biofilm. How this procedure was conducted?
Answer: This procedure is described in item 2.2.7 Antibiofilm activity of the M&M.
In all graphs you need control of the solvent of the PE
Answer: The solvent at the concentrations used showed no activity on the fungus, and was equal to the positive control. This information has been added in the results.
Line 305 EP?
Answer: A mistake occurred in the use of the abbreviation. We change in the final version of the manuscript.
Maybe add a picture of the biofilm/s
Answer: Thank you for the great suggestion, but unfortunately we didn't take enough quality pictures to put in the article.
Discussion
Explain why these two fungi where chosen especially the exophiala. Are they the main ethiological fungi in onychomycosis? This section in the discussion is not that convincing
Answer: First, with regard to superficial mycoses, the recurrence of mixed infections is underestimated, but cases of superficial mycoses from mixed infections are increasingly frequent in the literature. Furthermore, the etiopathogenesis of onychomycosis is closely related to the ability of its agents to form biofilms, but for a long time, the isolation of more than one species in nail scrape samples was erroneously considered to be culture contamination. However, data on the incidence of mixed infections in onychomycosis have been concerning. Thus, this case caught our attention because it is the first report in the literature of isolation of both yeasts, Candida parapsilosis and Exophiala dermatitidis causing onychomycosis. This information has been added in M&M, results and discussion.
Compare your results to other fungi as: Candida, Trichophyton
Answer: Thank you for the great suggestion, this information has been added in the discussion.
Compare the growth of the mixed biofilm with / without PE
Answer: Thank you for the great suggestion, this information has been added in the discussion.
The ability of the fungi to form biofilm should be understood in text that it is in microplates dishes not on biotics surfaces. The support surfaces in vivo are of quite different in chemical texture etc than your model.
Answer: Thank you for your observation. We completely agree. Thus, we put this information in the limitations of the paper.
Line 305 and 307: EP?
Answer: A mistake occurred in the use of the abbreviation. We change in the final version of the manuscript.
Line 271 change agents to microbes
Answer: Thank you for the great suggestion, we change to organisms in the final version of the manuscript.
271- on this section “to determine if the two agents are indeed causing onychomycosis” is not convincing. Use a biotic model.
Answer: Thank you for the great suggestion, we change this in the final version of the manuscript. It was not clear, but in the final version we emphasized that both C. parapsilosis and E. dermatitidis were isolated together from the same patient's toenail. We then evaluated in vitro the ability of these fungi to form biofilm separately, together, and in the presence of propolis extract. We highlight that to determine if two organisms are indeed causing onychomycosis, reproducibility of colony growth from the same collected sample is necessary. Thus for confirmation of the isolated agents, two independent collections were performed with an interval of one month after the first collection, repeating the same isolated agents in both collections.
You should aim the study at either proving that the two fungi are important etiological factors in onychomycosis or proving an effect of PE on fungi associated with the disease.
Answer: Thank you for the great suggestion, we change this in the final version of the manuscript and we have made the purpose of the article clearer.
Reviewer 2 Report
Reviewer comments :
Manuscript number : jof-2133003
The authors describe in this paper the effectiveness of propolis extract on mixed biofilm of Candida parapsilosis and Exophiala dermatitidis from onychomycosis.
The manuscript is well written and the figures are clear. Below some comments and corrections before the publication:
The title did not cover the first objective which was to assess the ability of these two species to form simple and mixed biofilm. It just mentions the effectiveness of the propolis extract.
What is the rationale of testing this extract on this mixed biofilm. It is common in real life?
Line 77: The authors stated they maintained the yeast in the mycotech. But they did not describe how (distilled water, glycerol…)
Line 80: They isolated and identified as C parapsilosis. This fungal species is part of the skin microbiota. How did the authors have incriminated C parasilosis as the pathogen.
Line 85: Which MALDI TOF? Bruker, Biomerieux…..It’s important to clarify
Line 90 -108: For these experiments, how did the authors have chosen the quantity of each solution used in the experiment. There is no reference of previous tests. Was it arbitrary choice, please clarify?
Line 171: How did they interpreted the Tukey’s test?
Line 175: Onychomycosis caused by the Candida genus is proximal in general. How did they explain the localization for this patient.
Author Response
Reviewer2
Manuscript number : jof-2133003
The authors describe in this paper the effectiveness of propolis extract on mixed biofilm of Candida parapsilosis and Exophiala dermatitidis from onychomycosis.
The manuscript is well written and the figures are clear. Below some comments and corrections before the publication:
The title did not cover the first objective which was to assess the ability of these two species to form simple and mixed biofilm. It just mentions the effectiveness of the propolis extract.
Answer: Thank you for the great suggestion, we change this in the final version of the manuscript to Mixed Infection of toenail caused by Candida parapsilosis and Exophiala dermatitidis and effectiveness of propolis extract on mixed biofilm.
What is the rationale of testing this extract on this mixed biofilm. It is common in real life?
Answer: Yes it is. Our research group showed the action of this propolis extract on fungi in biofilm (Tobaldini-Valerio et al., 2016) and for the treatment of onychomycosis (Veiga et al., 2018a, 2018b). Then, based on these studies, when the case of mixed infection arose, we decided to test the propolis extract to observe the action on this type of infection.
Tobaldini-Valerio, F.K.; Bonfim-Mendonça, P.S.; Rosseto, H.C.; Bruschi, M.L.; Henriques, M.; Negri, M.; Silva, S.; Svidzinski, T.I. Propolis: A Potential Natural Product to Fight Candida Species Infections. Future Microbiol. 2016, 11, 1035–1046.
Veiga, F.F.; Costa, M.I.; Cótica, É.S.K.; Svidzinski, T.I.E.; Negri, M. Propolis for the Treatment of Onychomycosis. Indian J. Dermatol. 2018a, 63, 515–517.
Veiga, F.F.; Gadelha, M.C.; da Silva, M.R.T.; Costa, M.I.; Kischkel, B.; de Castro-Hoshino, L.V.; Sato, F.; Baesso, M.L.; Voidaleski, M.F.; Vasconcellos-Pontello, V.; et al. Propolis Extract for Onychomycosis Topical Treatment: From Bench to Clinic. Front. Microbiol. 2018b, 9, 779.
Line 77: The authors stated they maintained the yeast in the mycotech. But they did not describe how (distilled water, glycerol…)
Answer: Thank you for your suggestion, we change this in the final version of the manuscript at M&M. "The yeasts were stored in Sabouraud Dextrose Broth (SDB; Difco™, USA) with glycerol at –80 °C."
Line 80: They isolated and identified as C parapsilosis. This fungal species is part of the skin microbiota. How did the authors have incriminated C parasilosis as the pathogen.
Answer: To determine if two agents are in fact causing onychomycosis, it is necessary that there is reproducibility of the growth of colonies from the same collected sample (Idris et al., 2019). In this case, the material collected from the nail showed scarce fungal structures in the direct mycological examination and in the culture in SDA there was growth of two yeasts concomitantly and reproducibly. Moreover, for confirmation of the isolated agents, two independent collections were performed with an interval of one month after the first collection, repeating the same isolated agents in both collections. For clarification, we have added this information in the final version of the manuscript, under M&M, results and discussion.
Idris, N.F.B.; Huang, G.; Jia, Q.; Yuan, L.; Li, Y.; Tu, Z. Mixed Infection of Toe Nail Caused by Trichosporon Asahii and Rhodotorula Mucilaginosa. Mycopathologia 2019, doi:10.1007/s11046-019-00406-y.
Line 85: Which MALDI TOF? Bruker, Biomerieux…..It’s important to clarify
Answer: For the MALDI TOF-MS method, the yeasts were prepared and processed with a Vitek MS mass spectrometer using the Myla or Saramis software for data interpretation. Thank you for the great suggestion, we added this information to the final version of the manuscript.
Line 90 -108: For these experiments, how did the authors have chosen the quantity of each solution used in the experiment. There is no reference of previous tests. Was it arbitrary choice, please clarify?
Answer: According to the studies performed by our biofilm formation research group, we have been adapting the quantities of solutions to provide favorable conditions for satisfactory biofilm formation (Veiga et al., 2018; Jarros et al., 2020; Salvador et al., 2022; Corrêa et al., 2020).
Veiga, F.F.; Gadelha, M.C.; da Silva, M.R.T.; Costa, M.I.; Kischkel, B.; de Castro-Hoshino, L.V.; Sato, F.; Baesso, M.L.; Voidaleski, M.F.; Vasconcellos-Pontello, V.; et al. Propolis Extract for Onychomycosis Topical Treatment: From Bench to Clinic. Front. Microbiol. 2018, 9, 779.
Jarros, I.C.; Veiga, F.F.; Corrêa, J.L.; Barros, I.L.E.; Gadelha, M.C.; Voidaleski, M.F.; Pieralisi, N.; Pedroso, R.B.; Vicente, V.A.; Negri, M.; et al. Microbiological and Virulence Aspects of. EXCLI J. 2020, 19, 687–704.
Salvador, A.; Veiga, F.F.; Svidzinski, T.I.E.; Negri, M. In Vitro Ability of Fusarium Keratoplasticum to Form Biofilms in Venous Catheter. Microb. Pathog. 2022, 173, 105868.
Corrêa, J.L.; Veiga, F.F.; Jarros, I.C.; Costa, M.I.; Castilho, P.F.; de Oliveira, K.M.P.; Rosseto, H.C.; Bruschi, M.L.; Svidzinski, T.I.E.; Negri, M. Propolis Extract Has Bioactivity on the Wall and Cell Membrane of Candida Albicans. Journal of Ethnopharmacology 2020, 256, 112791.
Line 171: How did they interpreted the Tukey’s test?
Answer: Our interpretation of Tukey's test was based on the difference between the hours of incubation of the biofilm (24 h, 48 h, 72 h, and 96 h) comparing all the hours with each other and thus, in addition to obtaining the analysis that there is a difference between the hours from the ANOVA, also being able to indicate which hour is different, either significantly increasing or decreasing.
Line 175: Onychomycosis caused by the Candida genus is proximal in general. How did they explain the localization for this patient.
Answer: A study by (Park et al., 2011) shows a picture of onychomycosis caused by Exophiala dermatitidis, with linear longitudinal grooves. Since the patient in our study has a mixed infection of the fungus, it is possible that the form of onychomycosis has adapted to the two fungi found on the nail.
Park, K.Y.; Kim, H.K.; Suh, M.K.; Seo, S.J. Unusual Presentation of Onychomycosis Caused by Exophiala (Wangiella) Dermatitidis. Clin. Exp. Dermatol. 2011, 36, 418–419.
Reviewer 3 Report
A severe case of nail fungus can be painful and may cause permanent damage to human nails. And it may lead to other serious infections that spread beyond feet if someone has a suppressed immune system due to medication, diabetes or other conditions. So the study of onychomycosis is of great clinical importance, and biofilm is related with the persistence of infection and resistance to antifungal drugs. This study is well organized and detailed.
I just have one question, is there any chance to figure out the chemical structure of the obtained biofilm, l guess it may closely related with the mode of action.
Author Response
Reviewer 3
A severe case of nail fungus can be painful and may cause permanent damage to human nails. And it may lead to other serious infections that spread beyond feet if someone has a suppressed immune system due to medication, diabetes or other conditions. So the study of onychomycosis is of great clinical importance, and biofilm is related with the persistence of infection and resistance to antifungal drugs. This study is well organized and detailed.
I just have one question, is there any chance to figure out the chemical structure of the obtained biofilm, l guess it may closely related with the mode of action.
Answer: Dear reviewer it is a great question and we agree. However, usually these relationships and chemical structures are related to interactions within the biofilm such as communication between species, factors which in this work were the focus and thus not evaluated.
Reviewer 4 Report
Ref.: ID jof-2133003
Major questions
The manuscript authorised by Salvador. et al. has been carefully and extensively revised. The potential applications of natural propolis against biofilm formation in cases of onychomycosis may be relevant in the design of new therapies against pathogenic fungi. I am afraid, however, that the manuscript fails in important methodological questions and other crucial points of conductance and presentation. In addition, the data presented are inconsistent and the statistical significance doubtful without an appropriate interpretation, and do not allow to propose a clear-cut conclusion. Thus, I cannot recommend the paper for publication in J. Fungi.
Major questions
-There is a main concern on data reproducibility, a crucial matter in any scientific investigation. The two yeasts studied have been isolated from a patient of onychomycosis and identified, without any further characterization of their most remarkable physiological and biochemical features. Therefore, I cast many doubts regarding the comparative results that might be obtained with other different isolates.
It is essential to perform these assays with standard clinical strains, which are available in reputed hospitals and clinical laboratories.
-The protocol for antifungal activity determination displays a number of drawbacks: (i) A precise description of the “propolis extract” preparation is necessary. The meaning of dry residue and TPC in the process is unknown, (ii) our group has a certain experience on the fungicidal action of propolis. We have recorded substantial differences in activity among distinct propolis s well as in several batches from a specific propolis. Therefore, although a specific identification of the single active principles should be preferable, at least a homogenization of PE ensures a coherent repetition of the assays. In this way, the use of alcoholic fractions is customary; (iii) Established methods (EUCAST or CLSI) have to be chosen for MIC calculation instead of direct observation.
-The same flaws apply to the biofilm assays. (i) I am disinclined to believe the procedure followed to measure biofilm viability (CFU) is reliable. The total cell number constituting the biofilm is unknown and the scrape and further sonication can also affect the degree of cell viability, so that the CFU counting could display great variations; (ii) Moreover, the results displayed in Fig. 2 are compared themselves, but no control for reference has been included (e.g., time 0); (iii) Assuming the absence of external perturbations, is a mystery why the cell number of E. dermatitidis in the biofilms first increases (24-48h), then decreases (72h) and finally raised again (96h) (Fig. 5, l. 200; Fig. 2). (iv) The authors should also provide a convincing explanation to the strange fact that the maximum biomass in biofilm is coincident with the lower metabolic activity (72 h) (Fig. 2); albeit this result seems abolished in mixed biofilms (?) (Fig. 3); (v) I am unable to understand the statistical analysis followed here.
-C. parapsilosis apparently exerts an inhibitory effect on E. dermatitidis during the formation of mixed biofilms. If true, the total cells of the latter in a mature biofilm should be inferior, which is not the case (Fig. 2). I also wonder about the relevance of this competition in the patient of onychomycosis, who was the source of both yeasts.
-Regarding the effect of propolis on planktonic cells, the proposed values of MIC are higher compared with other yeasts. A more precise concentration should be calculated by quantitative measurements. It seems strange the same value of MIC for the two yeasts, bearing in mind that E. dermatitidis development is hampered by propolis. The conclusion that MIC and MFC is identical is also gratuitous; the authors should carry out the corresponding determinations. Furthermore, propolis contain other components beyond polyphenols (flavonoids, terpenoids…) and the results cannot be directly extrapolated.
-Relevant bibliography regarding the composition and antifungal properties of propolis has been ignored here. Furthermore, there is an excessive number of auto-citations.
Other points
-The general style and English language is inadequate and must be revised and improved. There are some imprecise phrases and redundant words (e.g. “…chronic infection caused by nail fungi, primarily caused by, p. 1, l. 23; this sentence makes nonsense: “when performing the susceptibility… p. 6;, l. 238; “to observe E. dermatitidis fell into disfavour…p. 6, l. 253).
-Since SDA medium admits several concentrations of dextrose, the formula used here must be detailed.
-The expression of CFU/cm2 is rather anomalous (units of capacity are more common).
-The Discussion is rather ambiguous and imprecise.
Author Response
Reviewer 4
Major questions
The manuscript authorised by Salvador. et al. has been carefully and extensively revised. The potential applications of natural propolis against biofilm formation in cases of onychomycosis may be relevant in the design of new therapies against pathogenic fungi. I am afraid, however, that the manuscript fails in important methodological questions and other crucial points of conductance and presentation. In addition, the data presented are inconsistent and the statistical significance doubtful without an appropriate interpretation, and do not allow to propose a clear-cut conclusion. Thus, I cannot recommend the paper for publication in J. Fungi.
Major questions
-There is a main concern on data reproducibility, a crucial matter in any scientific investigation. The two yeasts studied have been isolated from a patient of onychomycosis and identified, without any further characterization of their most remarkable physiological and biochemical features. Therefore, I cast many doubts regarding the comparative results that might be obtained with other different isolates.
It is essential to perform these assays with standard clinical strains, which are available in reputed hospitals and clinical laboratories.
Answer: Besides, this case happened in a single patient, caught our attention because it is the first report in the literature of isolation of both yeasts, Candida parapsilosis and Exophiala dermatitidis causing onychomycosis. In this case, the material collected from the nail showed scarce fungal structures in the direct mycological examination and in the culture in SDA there was growth of two yeasts concomitantly and reproducibly. About the characterization, as shown in Figure 1, the yeasts are distinct phenotypically, where E. dermatitidis is a melanized yeast presenting its colony black, whereas C. parapsilosis is a hyaline yeast, presenting its colony white. Also, the yeasts were identified according to their morphological and biochemical characteristics, as well as, the use of MALDI-TOF MS that is based on the detection of specific proteins released from microbial cells. Moreover, for confirmation of the isolated agents, two independent collections were performed with an interval of one month after the first collection, repeating the same isolated agents in both collections. For clarification, we have added this information in the final version of the manuscript, under M&M, results and discussion.
-The protocol for antifungal activity determination displays a number of drawbacks: (i) A precise description of the “propolis extract” preparation is necessary. The meaning of dry residue and TPC in the process is unknown, (ii) our group has a certain experience on the fungicidal action of propolis. We have recorded substantial differences in activity among distinct propolis s well as in several batches from a specific propolis. Therefore, although a specific identification of the single active principles should be preferable, at least a homogenization of PE ensures a coherent repetition of the assays. In this way, the use of alcoholic fractions is customary; (iii) Established methods (EUCAST or CLSI) have to be chosen for MIC calculation instead of direct observation.
Answer: Thank you for the great observation. In the final version of the manuscript we addressed all these questions by clarifying the information. We allocate a specific paragraph about the propolis extract used that was characterized by Corrêa et al., 2020, which is part of our research group.
Corrêa, J.L.; Veiga, F.F.; Jarros, I.C.; Costa, M.I.; Castilho, P.F.; de Oliveira, K.M.P.; Rosseto, H.C.; Bruschi, M.L.; Svidzinski, T.I.E.; Negri, M. Propolis Extract Has Bioactivity on the Wall and Cell Membrane of Candida Albicans. Journal of Ethnopharmacology 2020, 256, 112791.
-The same flaws apply to the biofilm assays. (i) I am disinclined to believe the procedure followed to measure biofilm viability (CFU) is reliable. The total cell number constituting the biofilm is unknown and the scrape and further sonication can also affect the degree of cell viability, so that the CFU counting could display great variations; (ii) Moreover, the results displayed in Fig. 2 are compared themselves, but no control for reference has been included (e.g., time 0); (iii) Assuming the absence of external perturbations, is a mystery why the cell number of E. dermatitidis in the biofilms first increases (24-48h), then decreases (72h) and finally raised again (96h) (Fig. 5, l. 200; Fig. 2). (iv) The authors should also provide a convincing explanation to the strange fact that the maximum biomass in biofilm is coincident with the lower metabolic activity (72 h) (Fig. 2); albeit this result seems abolished in mixed biofilms (?) (Fig. 3); (v) I am unable to understand the statistical analysis followed here.
Answer: Dear reviewer, thank you very much for the observations.
(I) Regarding the methodologies used to evaluate biofilms, there are several direct and indirect methods that have been used to quantify cells in biofilms. Direct counting methods allow the counting of cells that can be cultured, including plate counts. To determine the CFU of the mixed biofilm, we optimize the entire scraping and sonication process to have the best possible yield without losing possible adhered biofilm on the plate and also eventual cell death due to sonication (see supplemental material). Indirect measurement methods include the determination of total biomass (cells, matrix, hyphae, pseudo-hyphae) by crystal violet assay and the quantification of cell viability by tetrazolium salts such as XTT, for example. Thus both direct and indirect methods are complementary since they provide distinct information about biofilm characteristics and architecture. (II) As for time 0, we got the number of cells right by counting in a Newbauer chamber and plating the initial inoculum to confirm the number of initial cells. Thus, we certify the exact amount of inoculum at the beginning of the biofilm formation process. (III) This information was added in the final version. (IV) This information was added to the M&M and discussion in the final version. (V) Our interpretation of Tukey's test was based on the difference between the hours of incubation of the biofilm (24 h, 48 h, 72 h, and 96 h) comparing all the hours with each other and thus, in addition to obtaining the analysis that there is a difference between the hours from the ANOVA, also being able to indicate which hour is different, either significantly increasing or decreasing.
-C. parapsilosis apparently exerts an inhibitory effect on E. dermatitidis during the formation of mixed biofilms. If true, the total cells of the latter in a mature biofilm should be inferior, which is not the case (Fig. 2). I also wonder about the relevance of this competition in the patient of onychomycosis, who was the source of both yeasts.
Answer: When performing the mixed biofilm formation of the two yeasts, it could be seen that together they were also able to form biofilm over time (Fig. 3 and 4A). However, with time, the number of viable cells of C. parapsilosis was higher than E. dermatitidis, but this black yeast remained present in all the evaluated times. Only in the presence of propolis that the elimination of E. dermatitidis is observed (Figure 4B). Onychomycosis by C. parapsilosis is common, however by E. dermatitidis is rare. Possibly E. dermatitidis benefits from this mixed biofilm situation and can infect and remain in the host causing the infectious process.
-Regarding the effect of propolis on planktonic cells, the proposed values of MIC are higher compared with other yeasts. A more precise concentration should be calculated by quantitative measurements. It seems strange the same value of MIC for the two yeasts, bearing in mind that E. dermatitidis development is hampered by propolis. The conclusion that MIC and MFC is identical is also gratuitous; the authors should carry out the corresponding determinations. Furthermore, propolis contain other components beyond polyphenols (flavonoids, terpenoids…) and the results cannot be directly extrapolated.
Answer: Dear reviewer, thank you very much for the observations. This information was added in the final version.
-Relevant bibliography regarding the composition and antifungal properties of propolis has been ignored here. Furthermore, there is an excessive number of auto-citations.
Answer: Dear reviewer, thank you for your comments. We included important references and chose to keep the other articles in the group, as this work is a continuation of the studies carried out by our group both in relation to the application of PE for the treatment of onychomycosis and in mixed biofilms and onychomycosis.
Other points
-The general style and English language is inadequate and must be revised and improved. There are some imprecise phrases and redundant words (e.g. “…chronic infection caused by nail fungi, primarily caused by, p. 1, l. 23; this sentence makes nonsense: “when performing the susceptibility… p. 6;, l. 238; “to observe E. dermatitidis fell into disfavour…p. 6, l. 253).
Answer: Thank you for your recommendation, we will send it to a native English speaker for revision. We do not send this to corrections back before we return them because of the short notice.
-Since SDA medium admits several concentrations of dextrose, the formula used here must be detailed.
Answer: The medium Sabouraud dextrose agar was made according to the manufacturer's directions on the label (SDA; Kasvi, Italy).
-The expression of CFU/cm2 is rather anomalous (units of capacity are more common).
Answer: CFU values were standardized according to the well area of the microplate, so we use the expression CFU/cm2.
-The Discussion is rather ambiguous and imprecise.
Answer: Thank you for the great observation, we seek to improve it by following the suggetions indicated by all the reviewers.
Reviewer 5 Report
The manuscript "Effectiveness of propolis extract on mixed biofilm of Candida parapsilosis and Exophiala dermatitidis isolated from onychomycosis" presents interesting research results. Needs some minor tweaking before publishing.
Detailed notes:
Specific figures should be added in the abstract.
The introduction is well written, it contains the most important introductory information on the topic.
Chapter 2.1 - has genetic identification been carried out? Have the strains been deposited in publicly available culture collections?
How was propolis extract obtained?
The presentation of the results and the discussion are correct.
There is no research summary as a separate paragraph.
The authors correctly refer to the literature (mainly current publications).
Author Response
The manuscript "Effectiveness of propolis extract on mixed biofilm of Candida parapsilosis and Exophiala dermatitidis isolated from onychomycosis" presents interesting research results. Needs some minor tweaking before publishing.
Detailed notes:
Specific figures should be added in the abstract.
The introduction is well written, it contains the most important introductory information on the topic.
Chapter 2.1 - has genetic identification been carried out? Have the strains been deposited in publicly available culture collections?
How was propolis extract obtained?
Answer: Thank you for the great observation. In the final version of the manuscript we addressed all these questions by clarifying the information. The propolis extract we used was the same one that Corrêa et al. used and characterized in their study, and as described in the paper, it was obtained from beehives located in northern Paraná State, Brazil.
Corrêa, J.L.; Veiga, F.F.; Jarros, I.C.; Costa, M.I.; Castilho, P.F.; de Oliveira, K.M.P.; Rosseto, H.C.; Bruschi, M.L.; Svidzinski, T.I.E.; Negri, M. Propolis Extract Has Bioactivity on the Wall and Cell Membrane of Candida Albicans. Journal of Ethnopharmacology 2020, 256, 112791.
The presentation of the results and the discussion are correct.
There is no research summary as a separate paragraph.
The authors correctly refer to the literature (mainly current publications).
Round 2
Author Response
Reviewer 1
Title:
Not really reflecting the manuscript
Answer: Thank you for the great suggestion, we change in the final version of the manuscript. “Case of mixed infection of toenail caused by Candida parapsilosis and Exophiala dermatitidis and in vitro effectiveness of propolis extract on mixed biofilm”
M&M:
Again, why not use molecular biology for identifications of the isolated fungi?
“The identification method by MALDI-TOF mass spectrometry was then per- 360 formed and confirmed Candida parapsilosis sensu stricto and Exophiala dermatitidis” - Give the MALDI TOF spectrum of the fungi including references spectrum
This is a corner stone of your hypothesis
Answer: LEPAC is a clinical laboratory in which it performs laboratory mycological diagnosis in its routine. The standardized methodologies that we routinely use are the macro and micromorphological analysis and we confirmed them by MALDI-TOFF in another public laboratory that serves the state of Paraná. These methodologies have already been standardized and confirmed with blind samples, as quality control, in compliance with current legislation. For this reason, we do not carry out the diagnosis by molecular biology.
Results:
The solvents are most important. I still think that the data on the effect of solvents on the microbes should be in with specific Concentrions and description of the solvents including company etc.
Answer: Thank you for the great suggestion, we change in the final version of the manuscript.
Discussion:
Need to make a clear note that this is kind of a “case” report as the fungi were isolated from one patient only. Need to clearly sperate in methods, results and discussion between the clinical isolated fungi and the lab strain ones. Need to further discuss the results in the discussion section.
Answer: Thank you for the great suggestion, we change in the final version of the manuscript.
Line # 83 NOT CLEAR;
Answer: Thank you for the great suggestion, we change in the final version of the manuscript.
UFC/cm2???;
Answer: We determined the colony forming unit by area.
MIC? CIM?;
Answer: We corrected the abbreviation of minimum inhibitory concentration to MIC.
# 193 MFC of biofilm ?
Answer: We determined the minimum fungicidal concentration for planktonic cells and not for biofilm.
#213 give references, elaborate on the technique
Answer: It was performed according to Corrêa et al. [33].
2.1 title “case” ???
Answer: Thank you for the great suggestion, we change in the final version of the manuscript.
Reviewer 2 Report
No further comments. Thank you for the responses
Author Response
Thank you for reviewing our article.
Reviewer 4 Report
Although the article has been revised and the authors have prepared answers to my constructive queries, I am afraid the changes carried out are rather cursory without substantial corrections. The title, position of authors and reorganization of a Figure has been varied. However, new experimental work in support of the general conclusions remains absent.
On the other hand, some modifications have hampered the general presentation, e.g. the substitution of MICs by CIMs is inappropriate, the Fig. 1 appears duplicated, and the alternative bibliography introduced does not apply to this specific approach.
Therefore, I am still reluctant and certainly think that rejection is in order.
Author Response
Reviewer 4
Although the article has been revised and the authors have prepared answers to my constructive queries, I am afraid the changes carried out are rather cursory without substantial corrections. The title, position of authors and reorganization of a Figure has been varied. However, new experimental work in support of the general conclusions remains absent.
On the other hand, some modifications have hampered the general presentation, e.g. the substitution of MICs by CIMs is inappropriate, the Fig. 1 appears duplicated, and the alternative bibliography introduced does not apply to this specific approach.
Answer: Thank you for your observation. Mistakes have been corrected and references have been checked.
Therefore, I am still reluctant and certainly think that rejection is in order.